

# Input Torque Measurements for Wind Turbine Gearboxes Using Fiber Optical Strain Sensors

Unai Gutierrez Santiago[1,2], Alfredo Fernández Sisón[2], Henk Polinder[1], and Jan-Willem van Wingerden[1]

[1]TU Delft 3mE, Mekelweg 2, 2628 CD, Delft, The Netherlands
[2]Siemens Gamesa Renewable Energy, Parque Tecnológico de Bizkaia, 48170 Zamudio, Spain

**Correspondence:** Unai Gutierrez Santiago (u.gutierrezsantiago@tudelft.nl)

**Abstract.** Accurate knowledge of the input torque in wind turbine gearboxes is key to improving their reliability. Traditionally, rotor torque is measured using strain gauges bonded to the shaft. Transferring the resulting signal from the rotating shaft to a stationary data acquisition system while powering the sensing devices is complex and costly. The magnitude of the torques involved in wind turbine gearboxes and the high stiffness of the input shaft pose additional difficulties. This paper presents a
new alternative method to measure the input torque in wind turbine gearboxes based on deformation measurements of the static first stage ring gear. We have measured deformation using fiber optic strain sensors based on fiber Bragg gratings because of their advantages compared to conventional electrical strain gauges. The present study was conducted on a Siemens Gamesa Renewable Energy gearbox with a rated power of 6MW, in which a total of 54 fiber optic strain sensors were installed on the outer surface of the first stage ring gear. The gear mesh forces between the planets and the ring gear cause measurable
deformations on the outer surface of the stationary ring gear. The measured strains exhibit a dynamic behavior. The strain values change depending on the relative position of the strain sensors to the planet gears, the instantaneous variations of the input torque, and the way load is shared between planets. A satisfactory correlation has been found between the strain signals measured on the static ring gear and torque. Two signal processing strategies are presented in this paper. The first procedure is based on the peak-to-peak strain values computed for the gear mesh events, and therefore, torque can only be estimated when a
gear mesh event is detected. The second signal processing procedure combines the strain signals from different sensors using a Coleman coordinate transformation and tracks the magnitude of the fifth harmonic component. With this second procedure, it is possible to estimate torque whenever strain data of all sensors is available, leading to an improved frequency resolution up to the sampling frequency used to acquire strain data. The method presented in this paper could make measuring gearbox torque more cost-effective, which would facilitate its adoption in serial wind turbines and enable novel data-driven control strategies,
as well as a more accurate assessment of the consumed fatigue life of the gearboxes throughout their operation.

## 1 Introduction

Scientists have long been warning us that we face a significant challenge regarding our climate. Renewable energies can play a pivotal role in reducing carbon emissions to enable a sustainable future. In the last decades, wind energy has seen a remarkable growth. Wind energy is already the second source of power generation in the EU when ranked according to installed capacity





(WindEurope.org, 2019). In Europe (EU27 + UK), the total installed capacity of wind power reached 220 GW in 2020, of which 194 GW is onshore wind. With 458 TWh generated, wind power covered 16% of Europe's electricity demand in 2020 (WindEurope.org, 2021). It is essential to reduce the levelized Cost of Energy (lCoE) from wind to guarantee further deployment of wind turbines towards the needed energy transition. Therefore, the lCoE has become one of the critical drivers for wind energy research in industry and academia.

In onshore wind energy, 75% of turbines have a geared drivetrain (van Kuik et al., 2016). The gearbox transfers the torque generated by the blades from the slow-speed rotor to the higher-speed generator. In the overall breakdown of costs, the gearbox is one of the main contributors because of the associated capital expenditure and the considerable contribution to operation and maintenance costs (Stehly et al., 2018). Gearbox reliability is improving, but gearboxes continue to be the largest wind turbine downtime source (Spinato et al., 2009) and generally do not reach the desired design life of 20 years (Sheng, 2013; Musial

et al., 2007). Therefore, improving gearbox reliability would lead to a significant reduction in the lCoE from the wind.

Drive train subsystem interactions and the effect of control strategy on gearbox loading are not fully known (van Kuik et al., 2016). The flexibility of the gearbox components influences the drivetrain's dynamic behavior and, therefore, the loading (Helsen et al., 2011). Traditional engineering models used for the simulation of gearbox input loads do not include this influence. Complex physically derived models have been built to study gearbox dynamics but have not been used together with

whole turbine models (Struggl et al., 2015; Girsang et al., 2014; Peeters et al., 2006). Current wind turbine design codes lack insight into the dynamic behavior of the internal drive train components. It is, therefore, highly desirable to be able to measure the torque from the rotor acting on the gearbox accurately and reliably. This torque will be referred to as the gearbox input torque.

The traditional method to measure torque is based on bonding strain gauges to the rotating shaft. The strain gauges convert

the deformation caused by the torque into a change in electrical resistance. Transferring the resulting signal from the rotating shaft to a stationary data logging system and powering the data acquisition devices is difficult and costly. In wind turbine gearboxes, the main shaft's deformation is small because of its high stiffness, which typically results in a low signal-to-noise ratio. These drawbacks have limited the use of such measurements to laboratory environments (Guo et al., 2017), validation and certification of experimental wind turbines, and troubleshooting exercises (Rosinski and Smurthwaite, 2010). More recently,

Zhang et al. (2018) explored different alternatives to measure torque in wind turbine drivetrains and added that a long-term measurement of torque is considered to be nonpractical or economically not feasible.

Fiber optic sensors have several advantages compared to electrical strain gauges (Kreuzer, 2006), the main ones for wind turbine applications are: 1) the signal to noise ratio of optical sensors is higher compared to conventional sensors, 2) they are immune to electromagnetic interference because they use light, and 3) many strain sensors can be accommodated in a single

fiber. Because of these qualities, fiber optic sensors have become popular in other wind turbine components. For example, fiber optic sensors are used in wind turbine blades for condition monitoring and design optimization purposes (Glavind et al., 2013).

The present paper develops a new method to measure the input torque of wind turbine gearboxes. The proposed method measures strain directly in the fixed frame. In this study, a total of 54 strain sensors were installed on the outer surface of the first stage ring gear. We have used fiber optic strain sensors to overcome the main limitations of electrical strain gauges.





The study has been conducted using a Siemens Gamesa Renewable Energy (SGRE) gearbox manufactured by Gamesa Energy Transmission (GET). The gearbox is a 3 stage gearbox, where the first and second are planetary stages, and the third is a parallel stage, with a rated power of 6 MW and a weight of approximately 44000 kg.

The main contributions of this paper are, 1) to develop a new method to measure input torque in wind turbine gearboxes based on deformations in the fixed frame, and therefore overcoming all the difficulties related to measuring on a rotating shaft,

2) the use of fiber optic strain sensors to explore their advantages compared to conventional electric strain gauges, 3) open the way to investigate the load sharing between planets, and 4) experimental demonstration of the above on a full-scale wind turbine gearbox.

The remainder of this paper is organized as follows, Section 2 gives a background on the fundamental principles used to measure torque from a static part in the gearbox, optical fiber sensing, and the test setup used for the experiments. Section 3

describes a signal processing procedure to estimate torque based on peak strain values of each individual sensor and Section 4 describes an alternative procedure based on a Coleman coordinate transformation to combine the instantaneous strain values of all sensors. Section 5 discusses the results obtained with both signal processing procedures. Finally, Section 6 draws the main conclusions of this work and suggests recommendations for future work.

## 2   Background

### 2.1   Gearbox fundamentals

The primary function of the gearbox is to transfer the torque generated by the wind from the rotor to the generator. In this transfer, the gearbox has to provide the needed increase in rotational speed. This speed increase is achieved in several stages. A variety of gearbox architectures have been used in commercial wind turbines, most of which combine planetary and parallel gear stages. Up to a rated power of around 2 MW, the most widely used configuration consist of an epicyclic planetary stage

followed by two helical parallel stages, also known in the industry as PHH or 1P2H. For more powerful turbines with larger rotor diameters (from around 3 MW to 8 MW), a gearbox configuration with two planetary stages and a single parallel stage has become dominant and is referred to as PPH or 2P1H.

The present study was conducted on a SGRE gearbox manufactured by GET with a rated power of 6 MW. This gearbox has a PPH configuration. A drawing of the shafts and gears in this gearbox is shown in Fig. 1. The structural housings of the

gearbox have been omitted for clarity. For even more powerful turbines, due to several factors like size constraints and the need to increase torque density, the industry is evolving to gearbox architectures with three planetary stages.

Regardless of the number of downstream planetary stages, the method presented in this paper can be applied to all gearbox configurations where the first stage is an epicyclic planetary stage with a stationary ring gear. In these gearboxes, the main shaft of the wind turbine is connected to the input or low-speed shaft (LSS) of the gearbox, which is the first stage planet carrier.

Figure 2 shows a section view of the first stage of the gearbox used in this study. When the planet carrier rotates, it transfers the input torque to the planet shafts. The planet wheels can rotate around the planet shafts and mesh simultaneously with the ring gear and the sun gear to achieve an increase in speed between the input (planet carrier) and the output (sun gear). The radial



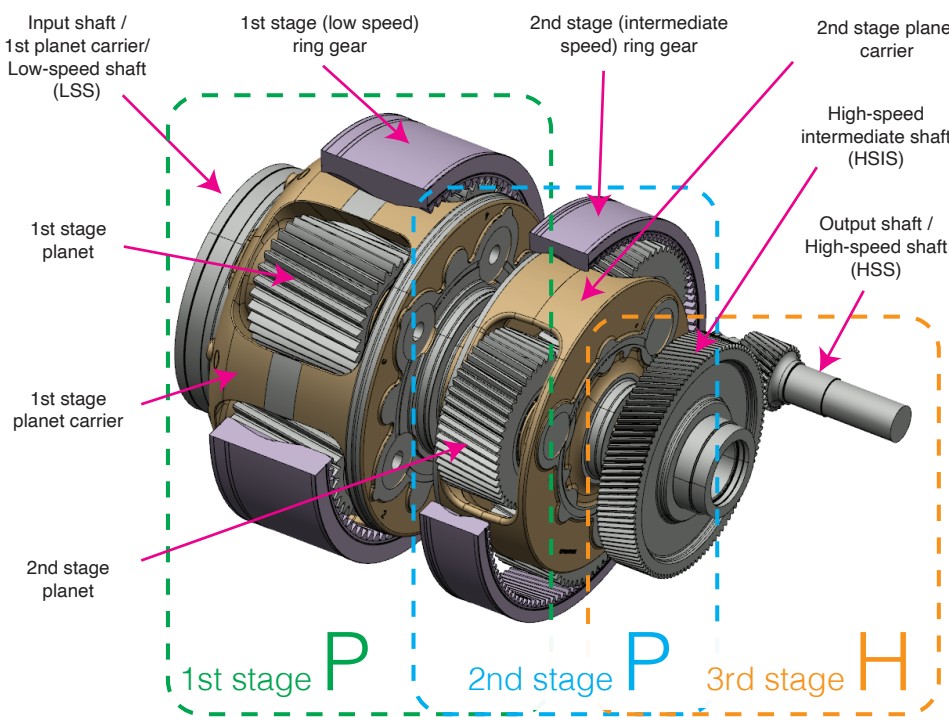

**Figure 1.** Assembly drawing of shafts and gears from the 3 stage Siemens Gamesa Renewable Energy gearbox (PPH configuration).

and tangential components of the mesh force acting from the planets to the ring gear are shown in Fig. 2, where they have been denoted as $F_{ti}$ for the tangential component and $F_{ri}$ for the radial component of the $i^{th}$ planet. The axial component of the gear mesh force has been omitted in this representation. These gear mesh forces deform the teeth of the ring gear and are then transmitted through the body of the ring to the reacting interfaces.

The first stage ring gear is the most expensive component of the gearbox in modern wind turbine gearboxes. Gearbox manufacturers strive to make the rim of the ring gear as thin as possible while complying with the minimum thickness requirements set by their design rules and gear rating standards like ISO 6336-1: 2019. The rim thickness for the first stage ring gear is defined as $S_R$ (see Fig. 2). Significant deformations are expected in the outer surface of the ring gear when the planets mesh with the ring gear because the rim is relatively thin. The research presented in this paper explores if and how strain measurements in the outer surface of the first ring gear can be used to derive the gearbox input torque.

## 2.2 Test bench set-up

Wind turbine gearboxes are typically tested in a back-to-back arrangement. Two gearboxes are connected through the low-speed shaft (LSS) to reproduce the torques generated by wind turbine rotors in a cost-effective manner. Figure 3 shows the



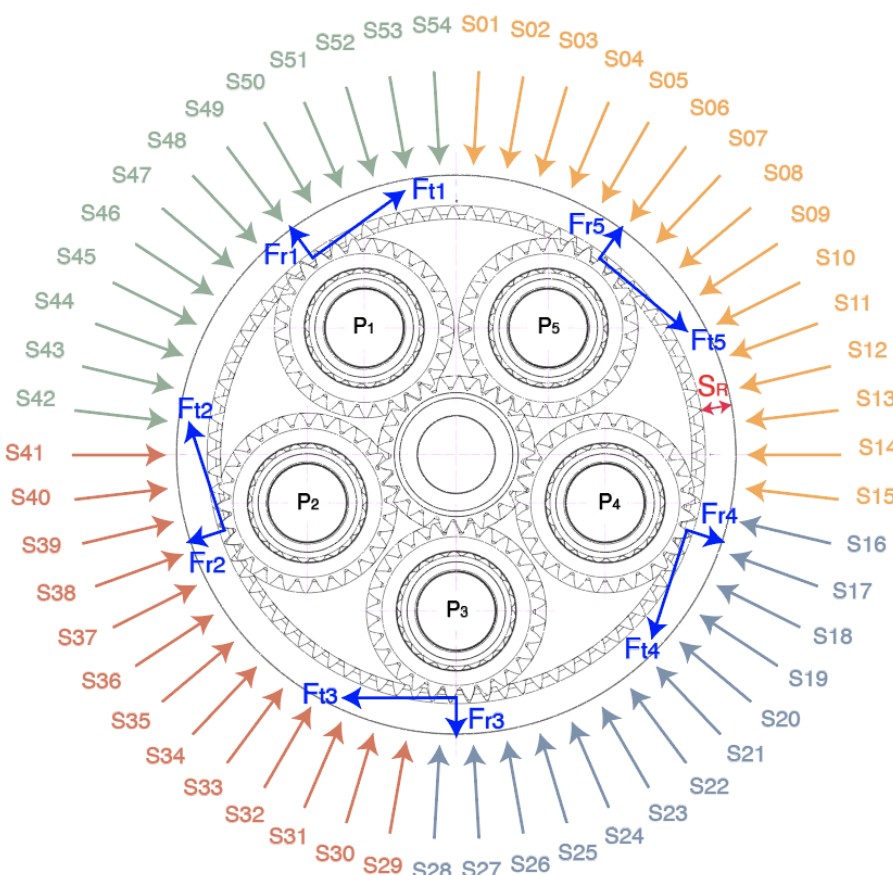

**Figure 2.** Rotor side view of the first planetary stage with the angular location of the strain sensors (S01 to S54). $S_R$ is the ring gear rim thickness and $F_{ti}$ and $F_{ri}$ are the tangential and radial gear mesh loads from the $i^{th}$ planet ($P_i$).

layout of the back-to-back test bench used for the present study, where electric motors produce the driving and braking torques. In the back-to-back configuration, a second gearbox is needed to reduce the rotating speed of the electric motor that acts as a driver and to increase the torque (referred to as "Test Gearbox 2" in Fig. 3). The input or driving motor is typically controlled to reach the desired running speed, and the driven motor is controlled to provide the specified braking torque, thus achieving the desired test conditions on "Test Gearbox 1".

Nevertheless, the gearbox operates at different boundary conditions compared to the wind turbine drivetrain configuration. The main three differences between a wind turbine and a back-to-back test bench are: 1) in a back-to-back test bench torque is the only controlled input excitation force; 2) the stiffness of the mechanical interfaces are different from the ones used in the nacelle of a wind turbine (mainframe, input or main shaft and HSS coupling); 3) there is a lack of a tilt angle in the test bench and the gearboxes are positioned with the main axis horizontally. Despite these differences, we consider the back-to-back test bench results representative of the wind turbine behavior when it comes to input torque.



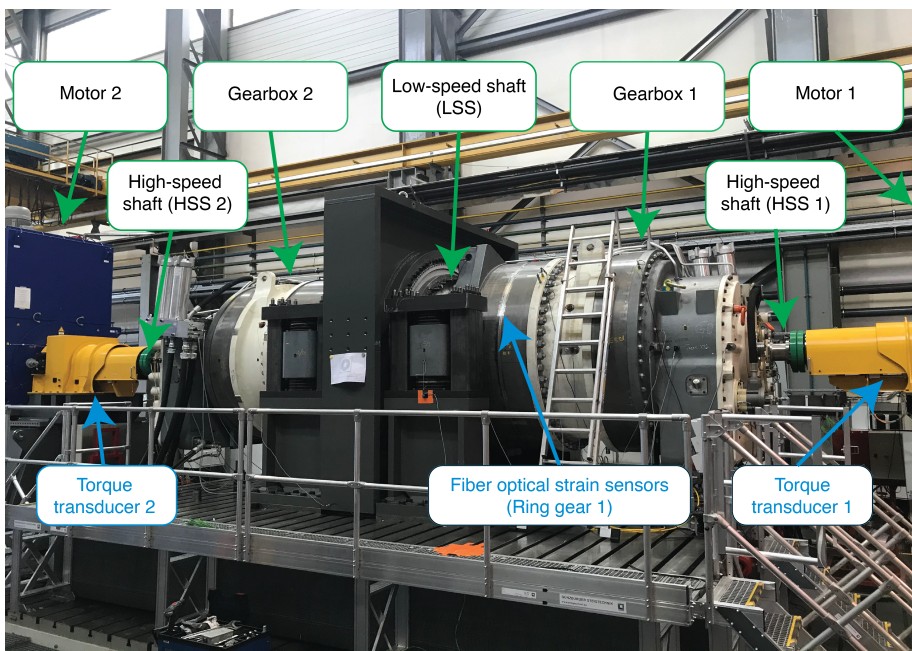

**Figure 3.** SGRE gearboxes on a back-to-back test bench (property of DMT GmbH & Co. KG). The first stage ring gear of "Gearbox 1" was instrumented with optical fibre strain sensors.

A full-scale gearbox with a rated nominal power of 6 MW was instrumented using optical fiber sensors. The position of the optical fiber sensors can be seen in Fig. 3 on the outer surface of the first stage ring gear of "Test Gearbox 1". All tests presented in this study were performed on the back-to-back test bench shown in Fig. 3, property of the company DMT GmbH & Co. KG (dmt group.com) at Krefeld (Germany) with electric motors of a rated power of 7.5 MW.

### 2.3 Fiber optical sensors

The fiber optical strain sensors used in the present study are based on Fiber Bragg Gratings (FBGs). These FBGs are a periodic variation of the refractive index of the fiber's core. At each refraction change, a small amount of light is reflected, and all the reflected light combines into one large reflection at the wavelength where the grating period satisfies the Bragg condition. The FBG is transparent for the light at wavelengths other than the grating wavelength, making it possible to integrate a large number of sensors in one fiber. The Bragg relation is:

$$\lambda_{\text{Refl}} = 2n\Delta, \tag{1}$$

with $n$ the index of refraction and $\Delta$ the period of the index of refraction of the FBG. The parameters $n$ and $\Delta$ depend on the temperature and strain at the grating. When a full spectrum is inserted into the fiber, a specific wavelength is reflected at each FBG sensor. The strain and temperature at each FBG can be determined by measuring the change in wavelength. Several FBGs can be integrated into a single optical fiber. In practice, the total amount of sensors per fiber is limited by the deformation to be





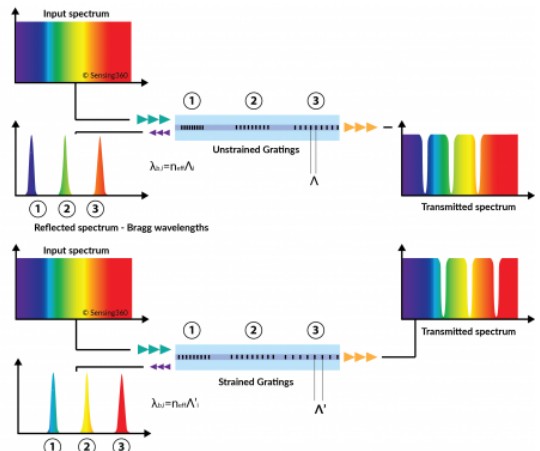

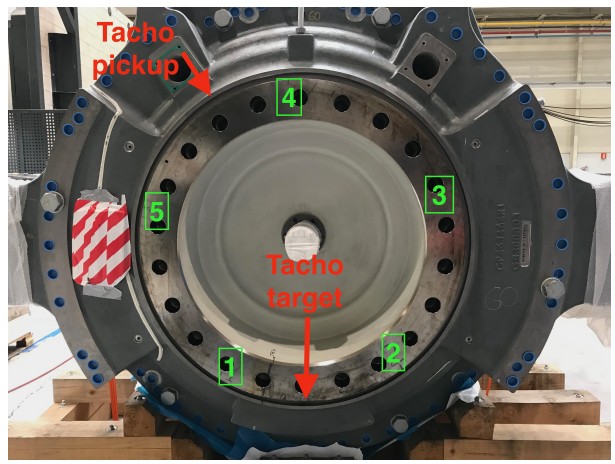

**Figure 4.** Working principle of Fibre Bragg Gratings used in fiber strain sensors, source Sensing 360 B.V..

**Figure 5.** Position of planets and input shaft marker tachometer (stationary inductive sensor and rotating target).

measured to ensure there is no overlapping between the wavelengths from different gratings. The working principle of FBGs is illustrated in Fig. 4.

For the present study, four optical fibres have been installed with 14 FBGs accommodated on each fiber. Out of the resulting

56 gratings, two were dedicated to temperature measurements and the remaining 54 gratings to measure strain. The four optical fibers were installed in the outer surface of the first stage ring gear, at the middle section as shown in Fig. 3. Figure 2 shows the radial and angular location of all the strain sensors with the corresponding labels in a rotor side section view. The four different colors for the sensor labels indicate how the FBGs were grouped into separate fibers (S01 to S15 in fiber number 1, S16 to S28 in fiber number 2, S29 to S41 in fiber number 3 and S42 to S54 in fiber number 4). The four optical fibers were connected

to an optical interrogator. The interrogator inserts a full spectrum of light into each fiber and acquires the reflected light to quantify the shift of wavelength from each grating. Three analog signals were added to synchronize the fiber optical strain measurements with the gearbox operating conditions: two torque transducers installed at the high-speed shaft (HSS) couplings (see Fig. 3) and an inductive sensor providing a once per revolution pulse of the input shaft (see Fig. 5). The purpose of this inductive sensor is to provide information on the angular position of the input shaft and, therefore, the relative position of the

planet carrier and the planets.

The fiber optical sensors were supplied and installed by the company Sensing360 B.V. (sensing360.com). Before installing the fibers, the outer surface of the ring gear was sanded to improve the bonding between the fiber and the ring gear. The fibers were glued to the ring gear outer surface using a cyanoacrylate adhesive. Figure 6 shows how the surface of the ring gear was prepared for the installation of the fibers. A naked optical fiber can be observed in Fig. 7 where two fiber gratings (S41 and

S42) illuminate when the fiber is exposed to a colored laser beam.




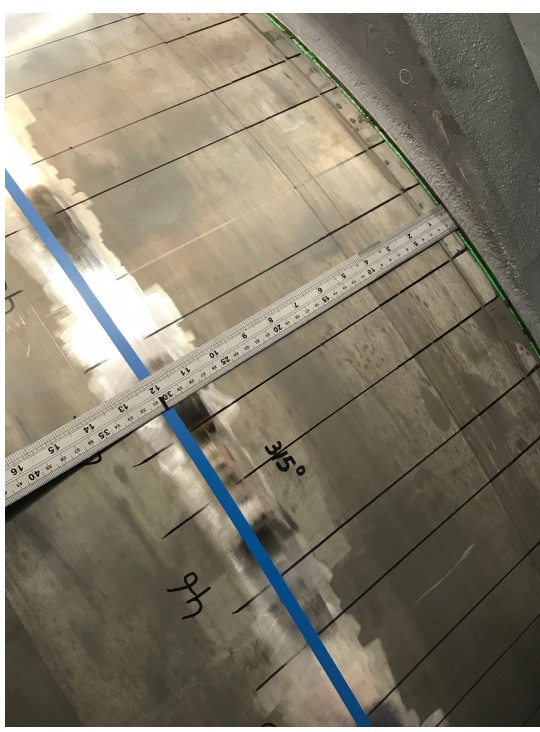

**Figure 6.** Ring gear outer surface preparation for fiber optical sensor installation

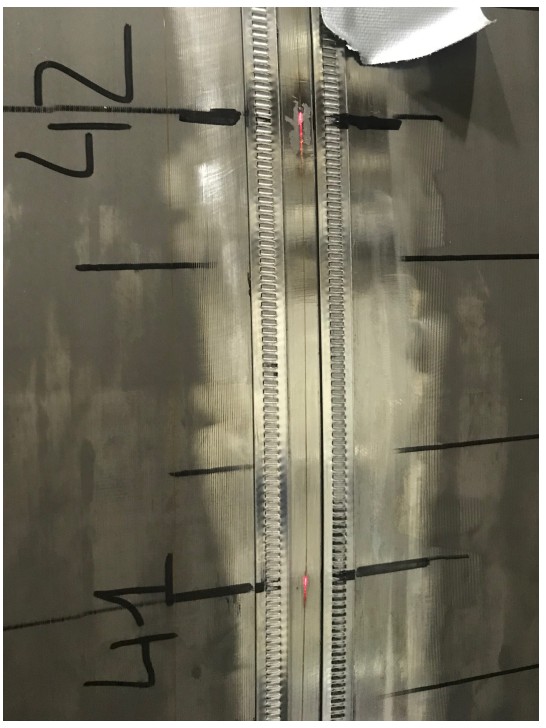

**Figure 7.** Detail view of optical fiber where two Bragg gratings are illuminated in red (S41 and S42).

## 2.4 Experiments

The results presented in this study were obtained during tests performed for the validation campaign of the SGRE gearboxes. In order to characterize the relationship between torque and strain, the instrumented gearbox was run under different stationary speed and torque conditions. During this torque vs. strain calibration process, 20 short tests were performed where the torque
level was increased in five percent increments from 5 to 100 % of the nominal, while the gearbox rotational speed was kept constant and equal to the nominal value.

After the stationary tests, the gearbox was run under dynamic torque conditions to collect strain data and evaluate the torque estimation procedures. In particular, two different tests with variable torque will be presented. First, a tests with a linearly increasing torque command. In both variable torque tests, the reference command speed of the test was kept constant. Then, a
second test where the torque level was changed in steps. Starting from a middle torque value, the torque command was changed to a lower and upper value. During both dynamic tests, the reference speed was kept constant.





## 2.5 Signal processing

Data from the optical fiber sensors were logged directly by an optical interrogator, and data from the analog signals were logged by a separate data acquisition system. Both systems are independent, but the data was time-stamped using the POSIX(R) (IEEE, 2008) time which allowed for time synchronization of sufficient accuracy. A sampling frequency of 2500 Hz was used to collect data from all 54 strain sensors.

Two preprocessing steps were applied to the signals from fiber optical sensors. First, the data was cleaned from glitches caused by the interrogator. A sudden drop in the value was observed in some of the logged signals. These drops were due to hardware communication errors and could be easily detected and removed. After the glitches in the data were removed, a moving average filter was applied. The resolution of each strain sensor is approximately 0.6 $\mu\epsilon$. Due to this limit in the combined resolution of the sensor and data acquisition system, a "staircase" effect was observed in the raw strain signals. The strain data signal can jump from one sample to the next in a set of fixed values separated by the sensor resolution. A moving average filter was applied to the 54 strain signals to filter this effect. The moving average was realized using a window size of 7 samples.

After the above mentioned preprocessing steps, strain signals were detrended to isolate the effect of changes in temperature to the measured signals, as explained in Section 2.3.

## 3 Torque estimation using peak-to-peak strain values

For each of the tests performed in the back-to-back test bench, described in Sections 2.2 and 2.4, data was acquired and logged from the 54 fiber optical strain sensors and the three analog signals described in Section 2.3.

### 3.1 Identification of strain peaks caused by gear mesh events

When the gearbox was tested under stationary torque and speed conditions, the individual signals from each fiber optical strain sensor exhibited a positive or tensile strain peak every time a planet meshed with the ring gear in the vicinity of the sensor. Figure 8 shows the acquired strain signal of the individual sensor "S01" during one revolution of the low-speed shaft. Each full revolution of the input shaft, is marked by pulse from the inductive sensor in orange, five positive or tensile strain peaks are observed corresponding to the meshing of the five planets in blue. A large tensile deformation occurs when a planet passes below the measurement location. As the low-speed shaft keeps turning, the strain diminishes, reaching compression. First, a local minimum is observed; then, the strain briefly recovers but drops again until a global minimum is reached. The strain keeps increasing from this global minimum until the next tensile peak.

In order to study the relationship between torque and strain, 20 tests were performed running the gearbox under stationary speed and torque conditions. While the gearbox rotational speed was kept constant and equal to the nominal speed, the torque was increased in five percent increments from 5 to 100%. In each test step, the strain signals of all 54 sensors were measured and logged for four minutes. The magnitude of the strain difference or peak-to-peak value between the detected maxima and

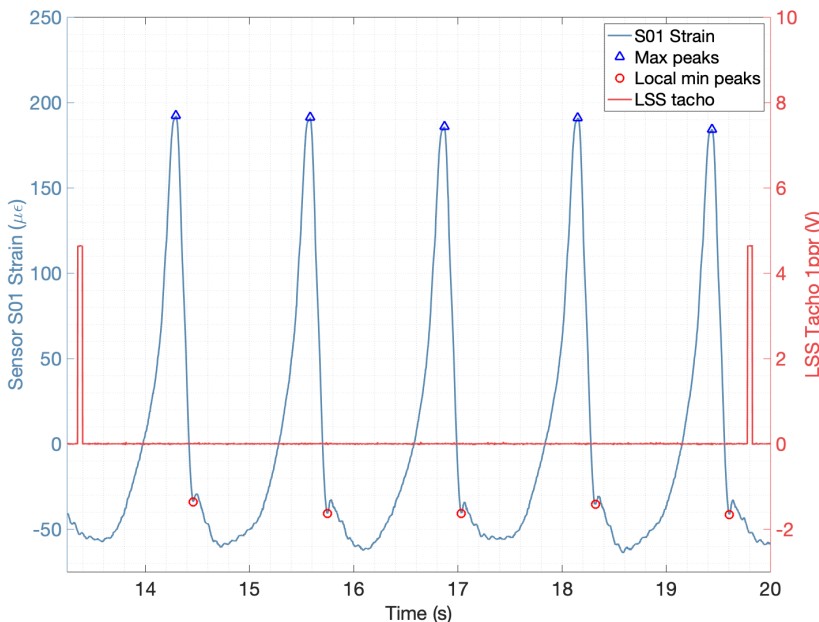

**Figure 8.** Sensor S01 Strain signal (left axis) during a single revolution of the input shaft (right axis) with detected peaks.

local minima peaks were computed for each of the 54 strain sensors for all twenty test steps. This magnitude will be referred to as the peak-to-peak value. The local minima were chosen because the time difference from the maxima to the corresponding

local minima is in the range of the gear mesh cycle and it is believed to be more representative of the gear mesh forces. Figure 9 shows the average peak-to-peak values computed for five tests at different torque levels using the local minima. The peak-to-peak value of each sensor is plotted according to the corresponding sensor angular location (rotor side view) shown in Fig. 2.

The input torque was not measured directly and was calculated using the torque data from the torque transducers installed in

the high-speed shafts (see Fig. 3). The efficiency of both gearboxes was assumed to be the same, and the LSS or input torque was computed as the average of both HSS torques multiplied by the gear ratio. The torque signals were averaged for the time segments between the tensile and compression peaks.

## 3.2 Relationship between peak-to-peak strain values and torque

Having measured strain on 54 locations on the outer surface of the static first stage ring gear, and with the simultaneous data

available from the torque transducers installed at both high-speed shafts, the following subsection shows how the measured strain signals can be correlated with the input torque.

Each measurement location has its own individual behavior with respect to torque, as can be seen in Fig. 9. A regression polynomial can be computed using the least-squares criterion for each individual strain sensor to fit the peak-to-peak values





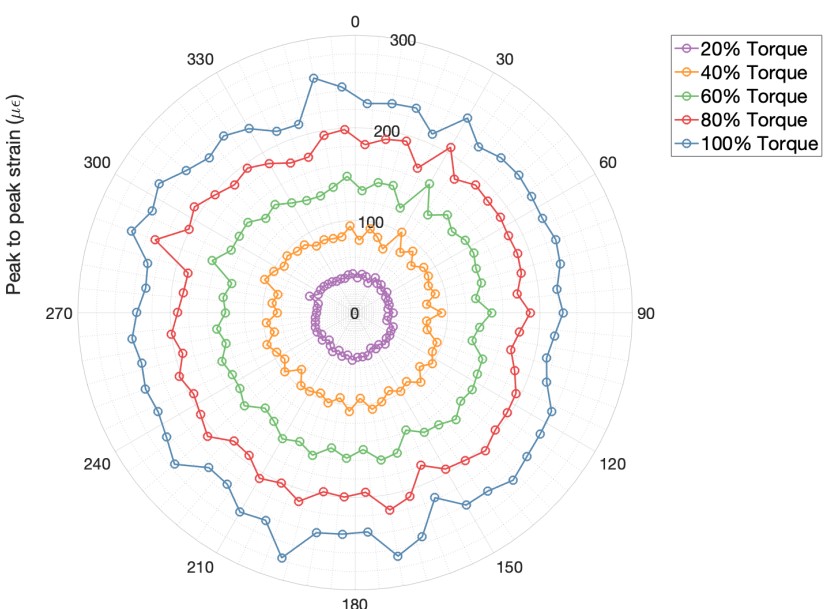

**Figure 9.** Average peak-to-peak strain values of all 54 sensors for five different torque levels (20, 40, 60, 80 and 100% torque).

with the torque. A linear, a quadratic, and a cubic fit of torque vs. peak-to-peak were investigated for comparison purposes.

Figure 10 shows the average peak-to-peak values vs. torque for three different measurement positions together with the cubic polynomial approximations. Only the the cubic regressions is shown for clarity purposes and similar regressions can be produced for every individual sensor as described in Section 3.1. When analyzing the different order regressions, the linearity of the peak-to-peak vs. toque behavior was observed to be different in each measurement location. The sensors S04, S17 and S53 were chosen for Fig. 10 because the difference in their behavior is representative of the largest differences observed. Although

a difference in stiffness was expected because of the lack of symmetry of the housings connected to the ring gear, these differences in linearity are not fully understood by the authors yet. Sensors S04, S17 and S53 were chosen for Fig. 10 because the difference in the peak-to-peak vs. toque behavior was more pronounced. Sensor S04 exhibited a lower more linear response while sensors S17 and S53 had a larger response but lower linearity. However, it is possible to achieve a satisfactory fit for all points by increasing the order of the regression polynomial.


### 3.3 Load sharing between planets

A polynomial fit of the average torque can be derived using the peak-to-peak values from each strain sensor. To extend this result for a dynamic situation, the load sharing between planets has to be known. The strain signals exhibit a highly dynamic

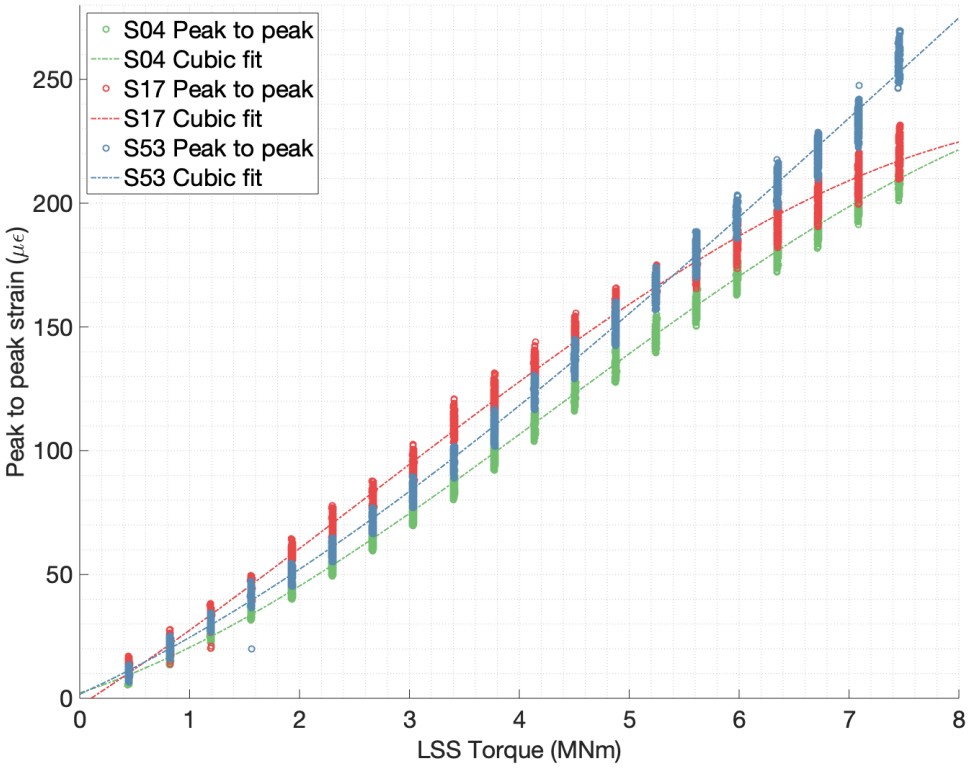

**Figure 10.** Peak-to-peak strain values of all detected gear mesh events of strain sensors S04, S17 and S53 vs torque in the low-speed shaft.

nature, and peak-to-peak values change with time. These changes can be due to changes in torque and possibly also due to
changes in load sharing between planets.

The gear rating standard ISO 6336-1 2019 defines the planet load share factor $K_\gamma$ as the load carried by the planet gear
carrying the higher load divided by the average load of all planets. Since a satisfactory fit has been found between the average
torque and the average peak-to-peak values, the following expression is proposed to derive $K_\gamma$ of a particular planet from the
measured peak-to-peak strain values

$$K_{\gamma i} = \frac{\overline{\Delta}_i}{\overline{\Delta}_{all}},\qquad(2)$$

where $\overline{\Delta}_i$ is the average peak-to-peak value from a particular planet $i$, and $\overline{\Delta}_{all}$ is the average peak-to-peak value of all planets.
It is possible to assign peak-to-peak values to individual planets because the position of the planets relative to the once-per-
revolution pulse is known (see Fig. 5). The meshing sequence of the planets is fixed if the sense of rotation is known. The
spacing between strain sensors is not an integer multiple of the number of planets, so with the instrumentation setup used in
this study, it is not possible to compare strain peaks in different positions simultaneously. In Fig. 11, the average peak-to-peak



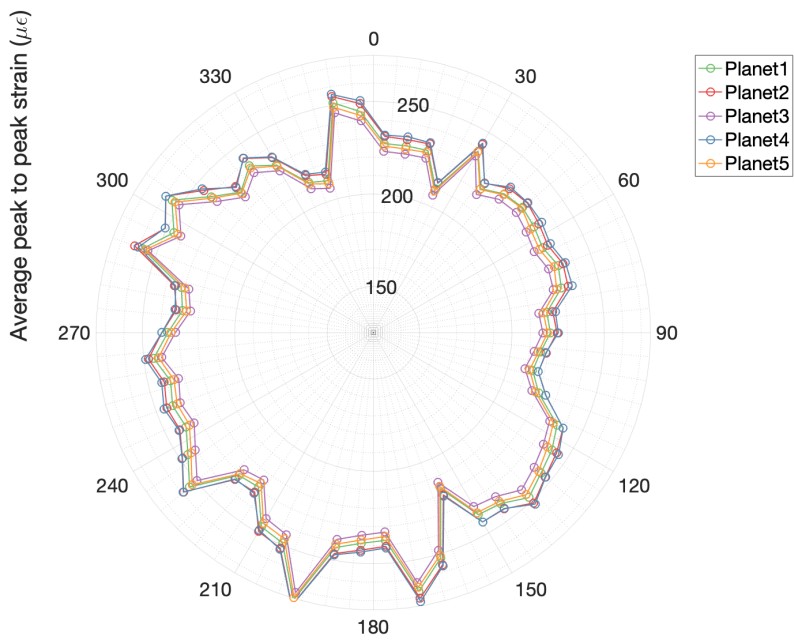

**Figure 11.** Average gear mesh peak-to-peak strain values for each planet.

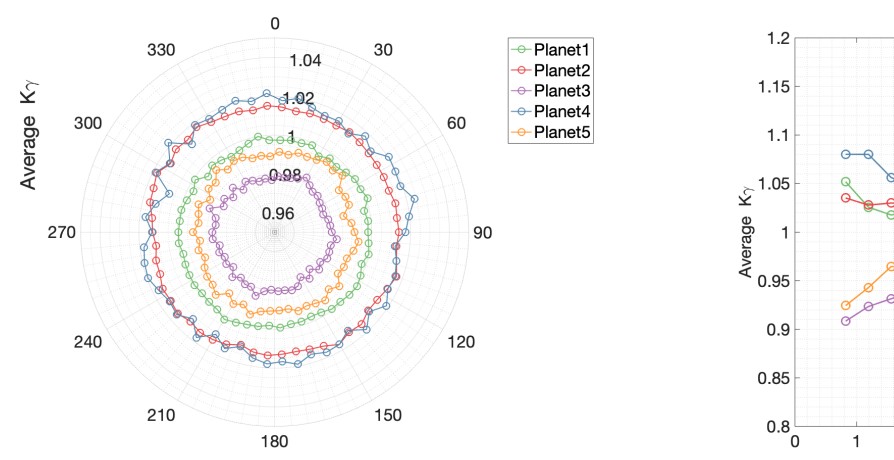

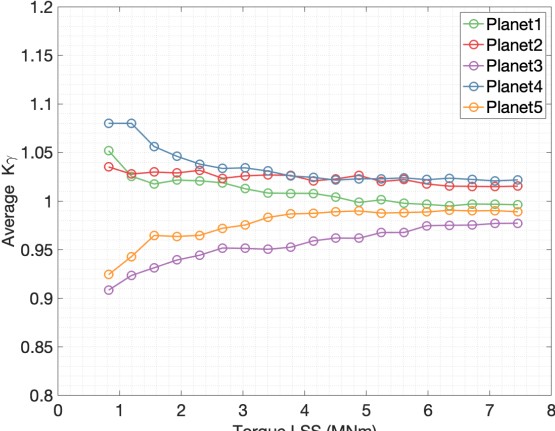

**Figure 12.** Average $K_\gamma$ for each sensor and planet.

**Figure 13.** Average $K_\gamma$ vs torque for the strain sensor S52.

values corresponding to each planet are shown in a polar plot for a test conducted at 100% torque. Figure 12 shows the resulting average $K_\gamma$ for each strain sensor position from the average peak-to-peak values shown in Fig. 11 obtained computing Eq. 2.



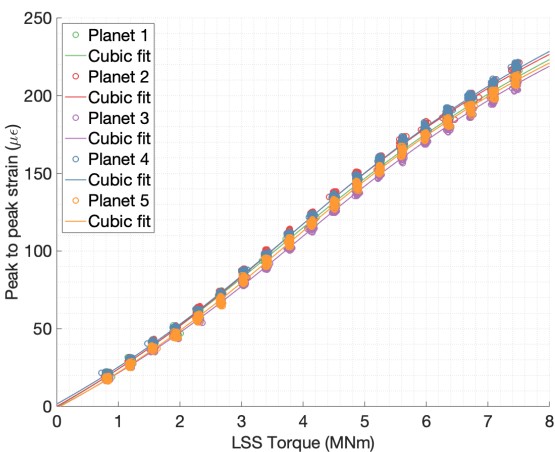

**Figure 14.** Peak-to-peak strain values of sensor S52 vs torque in the low-speed shaft separated for each planet and their corresponding cubic fit.

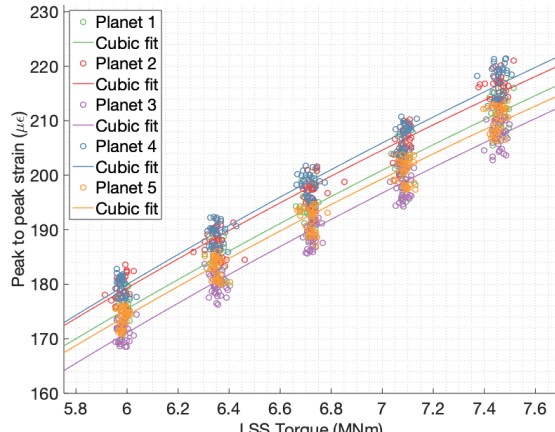

**Figure 15.** Peak-to-peak strain values of sensor S52 vs torque in low speed shaft separated for each planet (zoom for higher toque range).

Using the correlation of peak-to-peak values vs. torque and the average planet load share $K_\gamma$, an approximation of the instantaneous torque can be made. Figure 13 shows the average $K_\gamma$ values of the five planets for their corresponding average
torque values. As it can be seen in Fig. 13, the load sharing coefficient between planets depends on the gearbox input torque value. An improvement of $K_\gamma$ was observed, $K_\gamma$ closer to one, as the torque in the gearbox increases. In order to avoid a second linear regression, both data fitting steps can be merged into one. First, each detected mesh event is assigned to its corresponding planet. Then a regression polynomial is computed from the peak-to-peak strain values to torque for each individual planet.

### 3.4 Torque estimation procedure using peak-to-peak strain values

With the approximation to $K_\gamma$ presented in Section 3.3, torque can be estimated when a mesh event is detected. The procedure is represented in Fig. 16 and can be summarised as follows:

– Step 0 Calibration phase: Learn the peak-to-peak vs. torque correlation for each planet and each measurement location to produce $n * p$ regression polynomials, $n$ is the number of strain sensors and $p$ is the number of planets.

– Step 1 Detect peaks: for a new data set of strain signals, for every individual strain signal, detect gear mesh events and
identify their associated max and min peak strain values to compute the peak-to-peak value.

– Step 2 Assign planet: With the information from the once-per-revolution input shaft pulse, assign the detected gear mesh event to the corresponding planet that caused the strain.

– Step 3 Torque estimation: Evaluate the corresponding regression polynomial to compute a torque value from the peak-to-peak value (taking into account the strain sensor and the individual planet involved in the gear mesh).





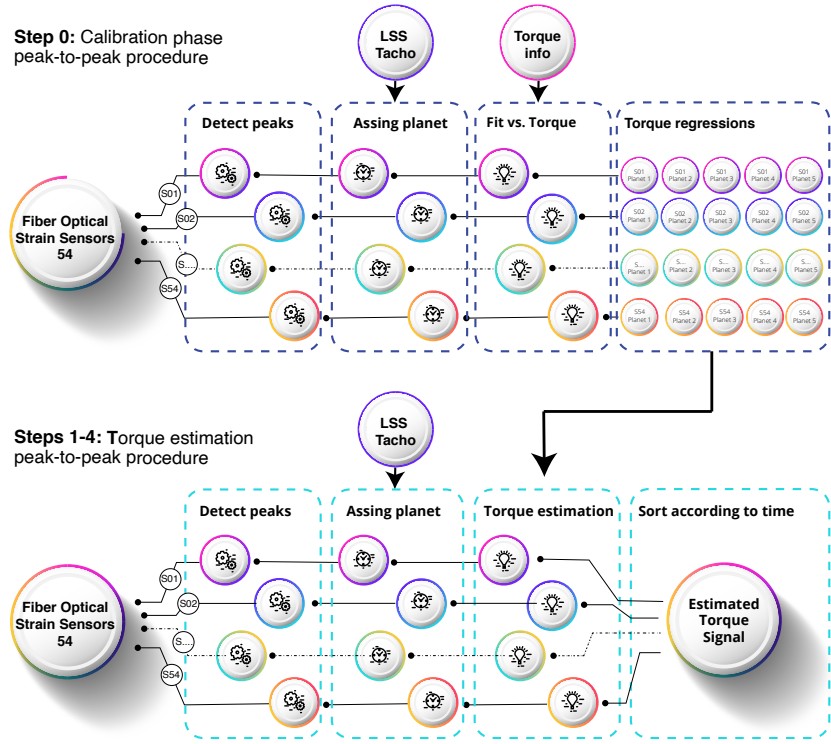

**Figure 16.** Torque estimation procedure based on peak-to-peak strain values.

– Step 4: Repeat steps 1 to 3 for all mesh events detected in all strain sensors and sort the estimated torque values from individual mesh events according to the time when the gear mesh event was detected.

## 4   Torque estimation using a coordinate transformation

With the data processing strategy for torque estimation based on peak-to-peak values presented in Section 3, torque can only be evaluated when a gear mesh event between a planet and the ring gear is detected. To overcome this limitation, an alternative 260  procedure has been developed based on a coordinate transformation of the strain signals, followed by an analysis and tracking of the harmonic components. This procedure combines the information from different strain sensors to exploits the signal information between mesh events.

The Coleman transformation or Fourier coordinate transformation can transform the equations of motion from a rotating coordinate system to a non-rotating coordinate system. In wind turbines, the Coleman transformation is also referred to as the 265  Multiblade Coordinate Transformation (MBC). The term MBC was adopted from helicopter theory (Johnson, 1994) and is widely used to analyze the dynamics of the wind turbine rotors (Bir, 2008; van Solingen and van Wingerden, 2015).



For the instrumentation setup used in the present study with 54 strain sensors, the Coleman transformation can be particularized to the following equations:

$$p_{nc}(t) = \frac{2}{54} \sum_{s=1}^{54} \delta_s(t) \cos\left(n\psi_s(t)\right),$$ (3)

$$p_{ns}(t) = \frac{2}{54} \sum_{s=1}^{54} \delta_s(t) \sin\left(n\psi_s(t)\right),$$ (4)

where $\delta_s(t)$ is the strain of the $s^{th}$ FBG sensor for a given time $t$ and $\psi_s(t)$ is the assigned angular position of the sensor at that time $t$ (see Section 2.3). The angle $\psi_s(t)$ is defined as the relative angle between the angular location of each sensor according to Fig. 2 and the angular location of the input shaft. The angular location of the input shaft determines the angular position of the planet carrier and, therefore, the five planets. The angular location of the input shaft is also known as the azimuth angle of
the rotor. The coefficients $P_{nc}(t)$ and $P_{ns}(t)$ can be computed for any integer multiple of $n$ and for any time $t$ when strain data of all sensors are available.

For low values of $n$, where the Nyquist criterion is satisfied, the expressions in Eqs. (3) and (4) are equivalent to the sine and cosine coefficients of a Fourier Series decomposition of the resulting periodic function $\delta(\psi)$. The function $\delta(\psi)$ is a discrete representation of the strain of the 54 measurement points $\delta$ as a function of their assigned angle $\psi$. Figure 17 shows the signal
values $\delta_s$ for all 54 strain sensors against their assigned angle $\psi_s$ for a given time sample $t_0$. In this case, $t_0$ corresponds to a time sample when the start of the once-per-revolution pulse was detected. The strain values of tests with different torque levels are shown in Fig. 17 using strain data samples that correspond to the same angular position.

The 54 strain sensors are equally spaced around the full revolution of the ring gear, and therefore, the function $\delta(\psi)$ is a periodic function with a period of $2\pi$. It's Fourier Series decomposition is defined as a sum of the basis functions $\phi_n(\psi)$ times
complex coefficients $p_n$:

$$\delta(\psi) = \sum_{n=-\infty}^{\infty} p_n \phi_n(\psi) = \sum_{n=-\infty}^{\infty} p_n e^{jn2\pi\psi} = p_0 + \sum_{n=1}^{\infty} (p_{nc} \cos n2\pi\psi + p_{ns} \sin n2\pi\psi)$$ (5)

In the field of rotating machinery, the frequency defined by the time needed to complete a full revolution of the shaft ($2\pi$ rad) is called the fundamental frequency, and the integer multiples of this frequency are called harmonic frequencies. The magnitude and phase of the complex coefficients $p_n$ can be computed as follows:

$$|p_n| = \sqrt{p_{nc}^2 + p_{ns}^2}$$ (6)

$$\angle p_n = \arctan\left(\frac{p_{ns}}{p_{nc}}\right)$$ (7)

It has been shown in Section 3.2 that each strain sensor has an individual relationship to torque. The Coleman transformation, Eqs. (3) and (4), combines the information from different strain sensors. A scaling or weighting factor was introduced to

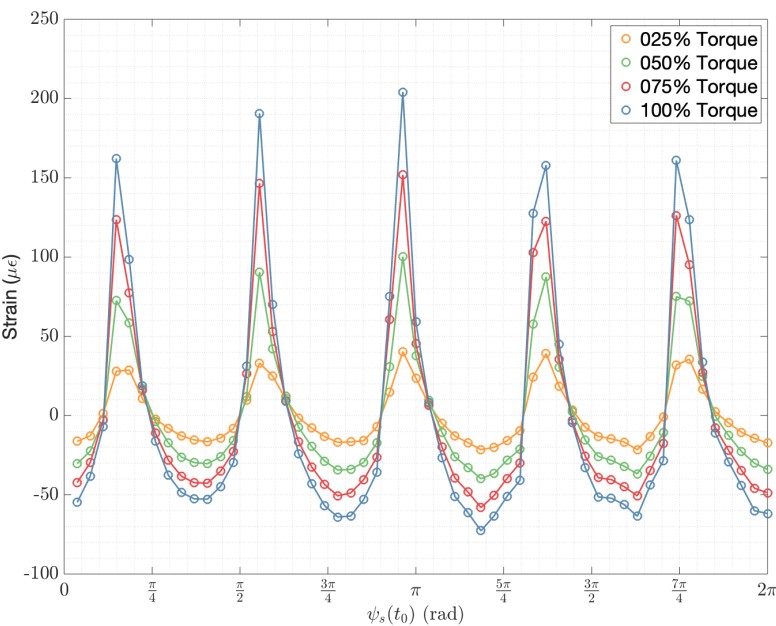

**Figure 17.** Instantaneous signal values of all 54 strain sensors for four different torque levels (25, 50, 75 and 100% Torque).

guarantee that all sensors have an equal contribution. To achieve these scaling factors the signals of each sensors were evaluated for complete revolutions of the input shaft. The strain signal from an individual strain sensor during a complete revolution of the input shaft is shown in Fig. 8. The power spectrum of an individual strain signal is dominated by the fifth harmonic component. This harmonic component can also be obtained from a Fourier Series decomposition if we consider $\delta_s(t)$ is a periodic function of time with fundamental frequency of $\omega_0 = \frac{2\pi}{T}$, where T is the time needed to complete a full rotation of the input shaft. The Fourier Series decomposition for the strain of each individual sensor as a function of time can be expressed as:

$$\delta_s(t) = \sum_{n=-\infty}^{\infty} a_{sn}\phi_{sn}(t) = \sum_{n=-\infty}^{\infty} a_{sn}e^{jn\omega_0 t} = a_{s0} + \sum_{n=1}^{\infty}(a_{snc}\cos n\omega_0 t + a_{sns}\sin n\omega_0 t) \tag{8}$$

The complex coefficients $a_{sn}$, are the $n^{th}$ harmonic component of the $s^{th}$ strain sensor. Figure 18 shows a comparison between the raw strain signal of a single FBG, sensor number 01 (S01), and the fifth harmonic component during a complete revolution of the low-speed input shaft. Two different torque levels (25 and 100%) have been plotted in Fig. 18 to illustrate the effect of torque. Higher integer multiples of the fifth harmonic have a lower but significant contribution. For the calibration phase, tests were performed at 20 different torque levels and the magnitude ($|a_{sn}|$) was computed for every full revolution measured using the following equation:

$$|a_{sn}| = \sqrt{a_{snc}^2 + a_{sns}^2} \tag{9}$$

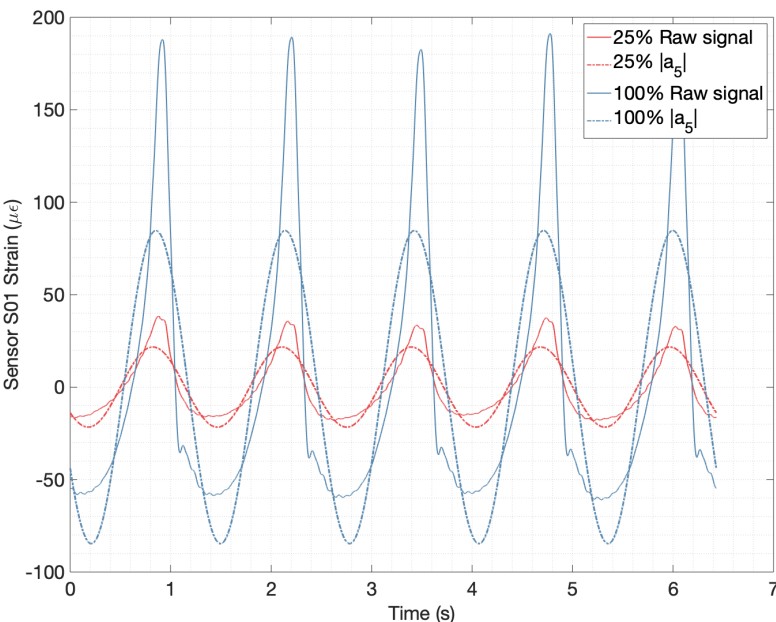

**Figure 18.** Original strain signal of sensor S01 for one shaft revolution vs reconstructed signal using the fifth harmonic component $a_5$ (two load cases are compared with 25% torque in red and 100% torque in blue).

The relationship between torque and the magnitude of the fifth harmonic of each sensor was investigated for all the 20 torque
levels. Again, the low-speed shaft torque value was derived from the measurements of the test bench torque transducers located
at the high-speed shafts. A regression polynomial was computed for each individual strain sensor fitting the $|a_5|$ values to the
torque using the leas-squares criterion. Figure 19 shows the $|a_5|$ magnitude values of four different FBGs, for all revolutions
of the input shaft measured during the 20 different load steps, against the average torque in the low-speed shaft during those
revolutions. A linear, a quadratic, and a cubic fit of torque vs. $|a_{s5}|$ for each sensor $s$ were realized to evaluate the linearity of
each sensor, however, for clarity only the cubic fit is shown in Fig. 19.

A good correlation was found for all 54 strain sensors between torque and the magnitude value of the fifth harmonic com-
ponent, and therefore, these $|a_{s5}|$ values were used to scale or weight the strain signals. For a full revolution of the input shaft,
all sensors witnessed the same torque and the signals of each strain sensors were scaled to have the same magnitude of the fifth
harmonic component. The weighting factors were computed using the following expression:

$$w_s = \frac{1}{54} \frac{\sum_{i=1}^{54} |a_{i5}|}{|a_{s5}|} \tag{10}$$

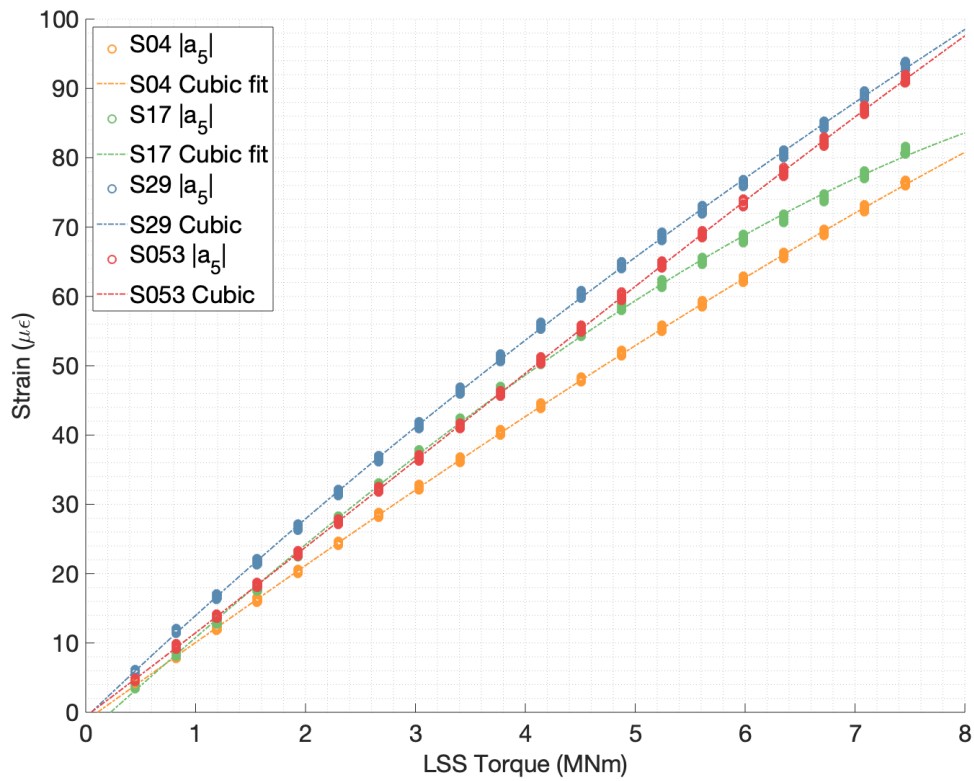

**Figure 19.** Magnitude of fifth harmonic $|a_5|$ of four strain sensors S04, S17, S29, S53 vs torque in low speed shaft.

Once every sensor is multiplied by its corresponding scaling factor, the weighted Coleman transformation can be written as follows:

$$p_{wnc}(t) = \frac{2}{54} \sum_{s=1}^{54} w_s \delta_s(t) \cos\left(n\psi_s(t)\right) \tag{11}$$

$$p_{wns}(t) = \frac{2}{54} \sum_{s=1}^{54} w_s \delta_s(t) \sin\left(n\psi_s(t)\right) \tag{12}$$

The magnitude and phase of the weighted complex coefficients $p_{wn}$ can then be computed with equivalent expressions to Eqs. (6) and (7) as follows:

$$|p_{wn}| = \sqrt{p_{wnc}^2 + p_{wns}^2} \tag{13}$$





$$\angle p_{wn} = \arctan\left(\frac{p_{wns}}{p_{wnc}}\right) \tag{14}$$

The once-per-revolution pulse signal is obtained from the inductive sensor shown in Fig. 5. The azimuth of the shaft in known when a pulse is detected but it can only be interpolated between pulses. The assigned angle $\psi_s$ was linearly interpolated between full revolutions of the input shaft. Attempts made to correlate the phase $\angle p_{wn}$ (Eq. 14) with torque were not successful. The accuracy of the angular position was not good enough to characterize the effect of torque on $\angle p_{wn}$.

The azimuth angle of the shaft is not strictly needed to compute the magnitudes of the harmonic components. Since the
54 strains sensors are equally spaced around the outer circumference of the ring gear, the relative angle between the angular location of the sensors and the azimuth of the shaft ($\psi_s(t)$) can be substituted in Eqs. (11) and (12) by the fixed angular location of each sensor, denoted as $\psi_{0s}$. Figure 20 shows the magnitudes of the $5^{th}$, $10^{th}$, $15^{th}$, and $20^{th}$ harmonic components ($|p_{w5}|$, $|p_{w10}|$, $|p_{w15}|$ and $|p_{w20}|$) plotted against the torque levels measured by the test bench torque transducers for all the revolutions measured during the calibration tests. A satisfactory correlation was found between the magnitude values of these harmonic
components and torque.

### 4.1   Torque estimation procedure with a coordinate transformation

A graphical representation of the procedure to estimate torque using the Coleman coordinate transformation and the $|p_{w5}|$ magnitude value of all 54 weighted sensors is shown in Fig. 21. The full procedure to estimate torque for every time sample can be summarized as follows:

– Step 0 Calibration phase: Perform tests with known torque to learn the strain vs. torque behavior. For each test, compute the magnitude of the 5th harmonic component $|a_{s5}|$ of all the individual strain signals. Compute the scaling factors $w_s$. Weight scale the strain sensors so that all sensors have an equal magnitude of the fifth harmonic component for a given torque. Combine the weighted strain values of all 54 sensors and compute the magnitude of the fifth harmonic component for each available time sample $|p_{w5}|$. Find a regression polynomial between LSS Torque and the magnitude
of fifth harmonic component of the combined strains.

– Step 1 Weight signals: For a new test where strain data has been logged, apply the weighting factor to all the strain sensors.

– Step 2 Coordinate transformation: For each time sample, apply the Coleman transformations to the weighted strain values to obtain $p_{wnc}(t)$ and $p_{wns}(t)$ using Eqs. (11) and (12) and compute the magnitude of the fifth harmonic component
$|p_{w5}(t)|$ using Eq. (13).

– Step 3 Torque estimation: Evaluate the regression polynomial to estimate torque from the combined magnitude of the fifth harmonic component of all 54 weighted instantaneous strain values.

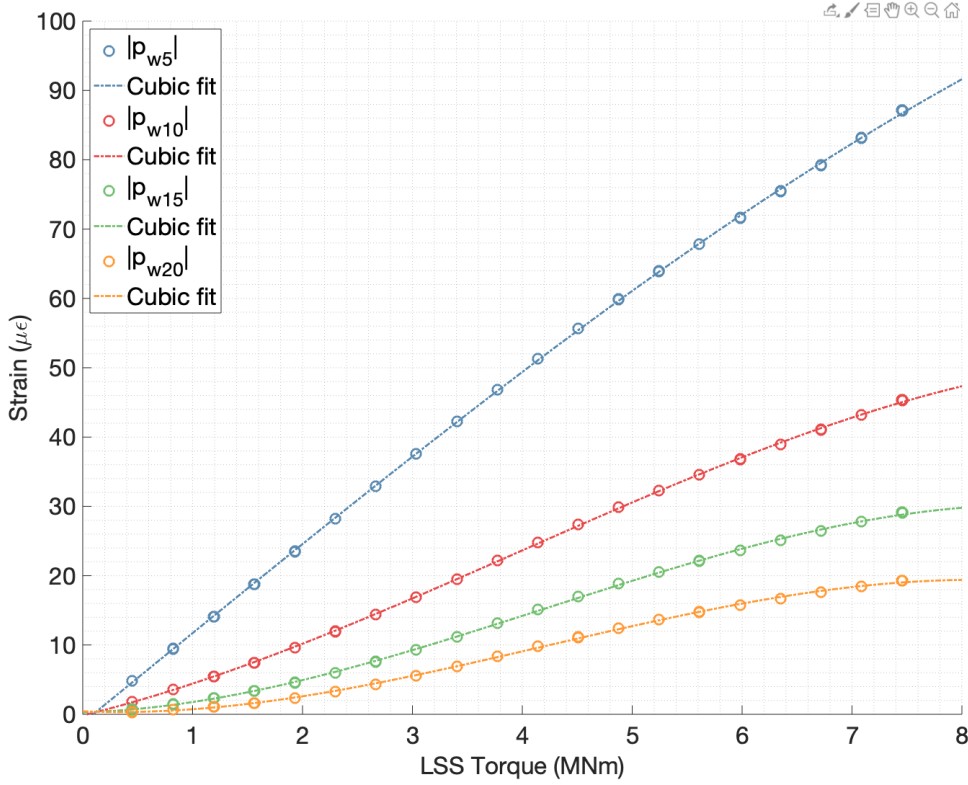

**Figure 20.** Magnitudes of the $5^{th}$, $10^{th}$, $15^{th}$, and $20^{th}$ harmonic components of all fifty-four weighted sensors vs torque in low-speed shaft.

## 5   Discussion

This section will discuss the main findings gathered during the development of the proposed method to measure input torque
in wind turbine gearboxes. Torque estimation results obtained using data from two dynamic load tests will be analyzed to
illustrate the key differences between the two alternative procedures presented in Sections 3 and 4.

Both procedures rely on a calibration phase, where the instrumented gearbox was operated under torque conditions known
from other measurements. In the present study, 20 short tests with different stationary reference torque levels were performed
to fit the strain to torque relationship (see Sections 3 and 4). Once the calibration phase had been accomplished, strain data
was collected for tests with dynamic torque conditions. Figure 22 shows the estimated torque measurements during a test with
a linearly increasing torque command. The torque estimate using peak-to-peak strain values is shown in blue, and the torque
estimate using the coordinate transformation is shown in red. Figure 23 shows the torque estimates achieved for a test where
the torque level was changed in steps. Starting from a middle value, the torque reference was changed to a lower and upper





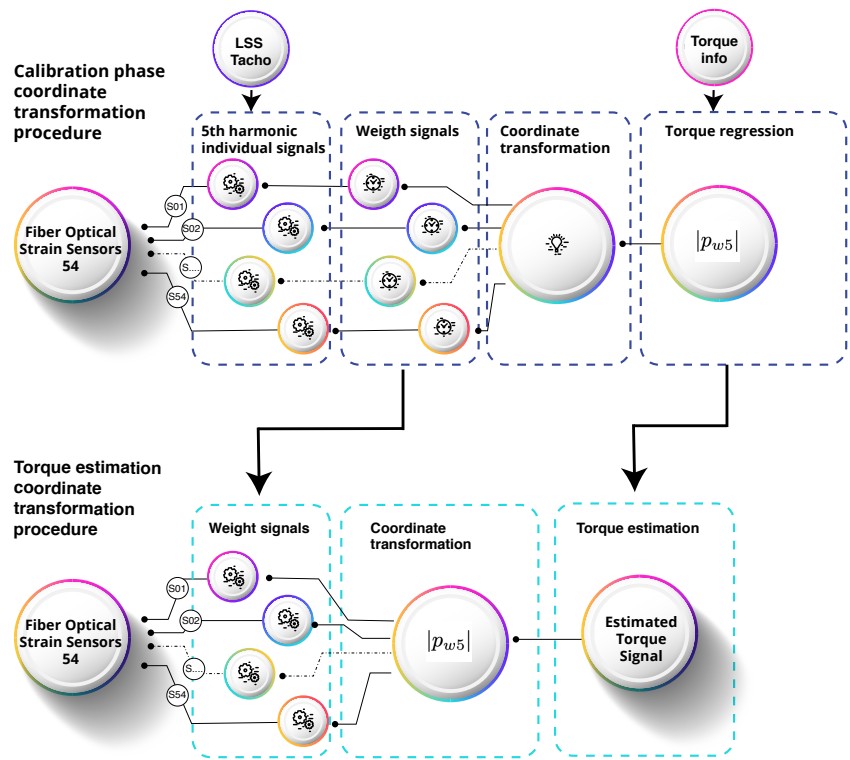

**Figure 21.** Torque estimation procedure based on a coordinate transformation.

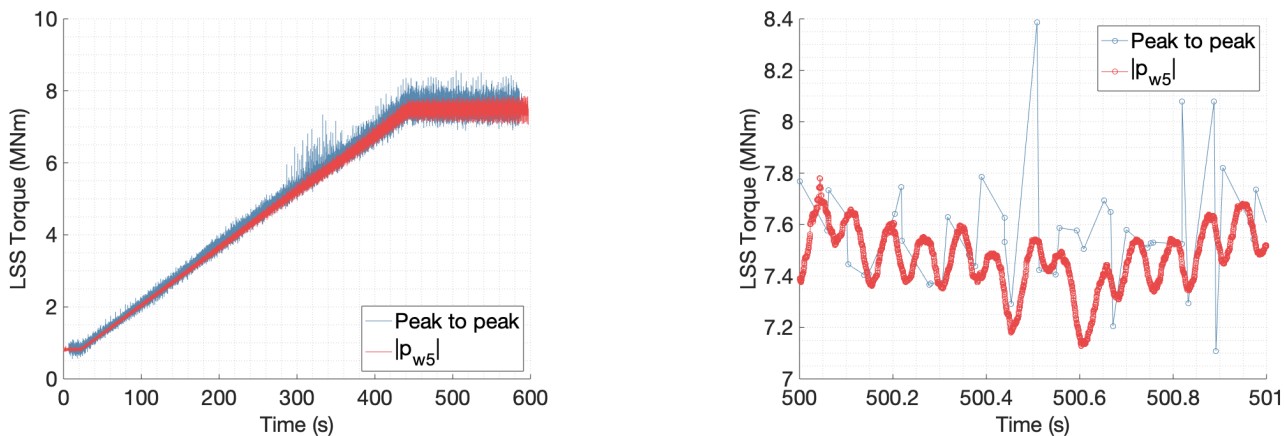

**Figure 22.** Torque estimates for a test with variable torque (ramp), results for entire test on the left and 1 second detail on the right.

value. During both dynamic tests, the reference speed was kept constant. For clarity, a detailed window of 1 second is shown

to compare the estimates of both tests.





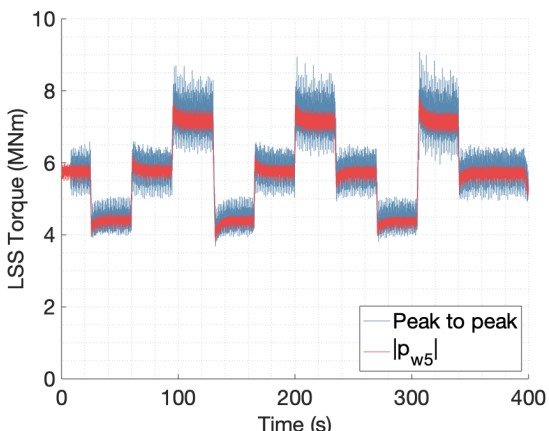 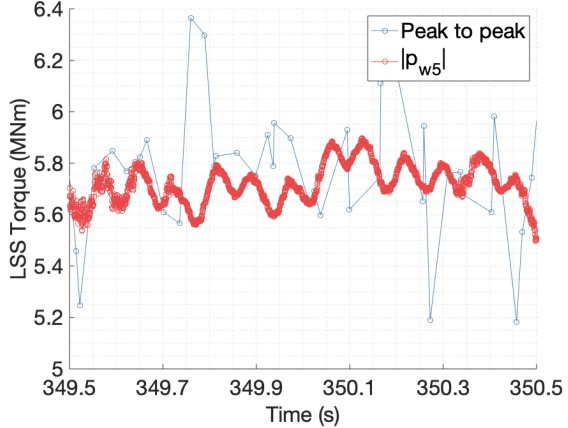

**Figure 23.** Torque estimates for a test with variable torque (steps), results for entire test on the left and 1 second detail on the right.

– Temporal resolution:

In the signal processing alternative based on peak values of strain, torque can only be evaluated when there is a gear mesh event between a planet and the ring. Therefore, the frequency resolution of the estimated torque depends on the gear mesh frequency of the first stage and the number of strain sensors used. The second signal processing strategy combines
the simultaneous information of different strain sensors using a coordinate transformation. This allows tracking the magnitude of the harmonic components of the combined weighted strain values. With a total of 54 strain sensors equally spaced around the first stage ring gear (see Fig. 2), the gearbox input torque can be estimated for each available time sample, which results in a frequency resolution equal to the sampling frequency used to acquire the strain data. The difference in frequency resolution of the estimated torque from both procedures can be seen in the one-second windows
of Fig. 22 and 23.

For future implementations of the method presented in this paper, it is possible to achieve a trade-off between spatial resolution and frequency resolution, selecting the sampling frequency and the number of sensors. If a lower number of strain sensors was to be used, for example, if we halve the number of sensors to 26, then by taking two consecutive strain samples of each sensor, we can still compute the magnitude of the fifth harmonic component and the resulting frequency
resolution of the torque estimate would be half the sampling frequency used to acquire strain data. If the number of sensors is low, the accuracy of the magnitude of the fifth harmonic component has to be considered. In the extreme, with a single strain sensor, it would only be possible to evaluate the input torque once every revolution, which would result in a significantly lower temporal resolution.

– Load sharing between planets (average and instantaneous $K_\gamma$):

Knowledge of the load sharing between planets is needed to estimate the dynamic instantaneous torque value in the procedure based on peak-to-peak strain values. An inductive sensor was added to the instrumentation to provide a once-





per-revolution pulse, making the mesh sequence of planets known. Therefore, it is possible to assign peak-to-peak values to individual planets. The load share between planets can be computed using Eq. (2). However, in the setup presented in this paper, the number of strain sensors is not an integer multiple of the number of planets. When a strain peak caused by the gear mesh of a planet is detected in a given strain sensor, the rest of the strain sensors do not observe the peaks caused by the mesh of the other planets simultaneously. Therefore, only an average $K_\gamma$ (planet load share factor) can be extracted for each measurement position, and the true instantaneous load sharing between planets could not be evaluated. The instantaneous torque value was approximated, assuming that the instantaneous value of $K_\gamma$ is equal to the average. A larger variation of torque was observed in the torque estimates produced by the peak-to-peak procedure shown in Figs. 22 and 23. The authors believe that the approximation made for $K_\gamma$ is the main reason to justify the larger torque variations. Since the average $K_\gamma$ is used, if the instantaneous load sharing between planets is worse than the average (higher instantaneous $K_\gamma$ values), the torque variations would be overestimated.

– Sensing of shaft's azimuth angle and 'real-time' capability:

The fact that a complete revolution of the input shaft has to be measured in the signal processing alternative based on peak-to-peak strain values to assign the suitable planet to the detected strain peaks is also responsible for the delay in the peak-to-peak torque estimates shown in Figs. 22 and 23. Torque can only be evaluated after the first pulse of the input shaft inductive sensor is detected when the peak-to-peak procedure is used. On the other hand, the procedure based on the coordinate transformation can be accomplished without the inductive sensor because the azimuth angle of the shaft is not needed to compute the magnitude of the harmonic components. This can simplify the instrumentation set-up and enable to estimate torque in "real-time".

– Weighting of individual strain sensors:

An important finding during the calibration phase of both processing strategies was that each measurement location showed a slightly different sensitivity to torque (see Figs. 10, 13, 17, and 20). Several factors can play a role in explaining this. On the one hand, installing the strain sensors is a manual process, and differences in how the fibers are glued to the surface of the ring gear and small positioning errors are expected. On the other hand, the ring gear is connected to structural gearbox components with highly asymmetric stiffness. This lack of symmetry is most pronounced for the FBG sensors close to the torque reaction arms in the housing on the rotor side of the ring gear, shown in Figs. 3 and 5. Because of the reasons mentioned above, differences in stiffness are expected. However, the difference in the linearity of the torque-strain relationship is not fully understood by the authors yet. In any case, it is possible to achieve a satisfactory fit for all measurement points by increasing the order of the regression polynomial, and therefore an accurate torque estimate can be achieved. To compensate for the different sensitivity of each measurement location, we have used the average magnitude of the fifth harmonic component of the individual strain signals as a basis for weighting the strain values. All 54 sensors are weighted to exhibit the same relationship to torque. This makes it possible to combine the sensors.





– Accuracy:

In the experimental setup used for this study, a direct torque measurement in the input shaft was not available, the torque transducers were installed in the high-speed shafts, and therefore a quantitative evaluation of the torque estimates could not be performed. For future work, a direct comparison between the method presented in this study and a calibrated direct measurement is suggested.

In the signal processing alternative based on the Coleman transformation, the weighted strain values of all the sensors are combined to compute the magnitude of the fifth harmonic component for each available time sample $|p_{w5}(t)|$. The torque is estimated using the regression polynomial between LSS Torque and $|p_{w5}|$. Figure 20 shows that higher-order harmonics (10, 15 and 20) can also be correlated with torque. The procedure presented in this paper could be extended to consider these magnitudes and combine them to achieve higher accuracy torque estimates. Again, a quantitative analysis

of the accuracy of the estimates could not be performed with the instrumentation set-up available and is recommended for future work.

Finally, it is worth noting that the potential effect that non-torque loads could have on the method presented in this study could not be evaluated. The test bench used to run the experiments, see Section 2.2, is a back-to-back gearbox test bench where only torque and speed can be controlled. The effect of these non-torque loads in four-point mount gearboxes is expected to

be small. However, to fully demonstrate the applicability of the presented methodology in a wind turbine installation, it is suggested to investigate the behavior of the presented method in a test bench with non-torque loading capabilities, i.e., axial forces and bending moments.

## 6    Conclusions

This paper develops a new method to measure the input rotor torque of wind turbine gearboxes. The proposed method is

based on strain measurements of the static first stage ring gear. Measuring in the static frame overcomes the main drawback of traditional methods, which measure the strain of rotating components. Additionally, optical fiber strain sensors were used because of their advantages over more conventional electrical strain gauges.

A satisfactory correlation was found between the strain signals measured on the static first stage ring gear and torque. Two signal processing strategies have been presented in this paper for determining input rotor torque. The first is based on the

peak-to-peak strain values assigned to the gear mesh events. The second is based on a coordinate transformation of all the strain signals, followed by tracking the magnitude of the fifth harmonic component. The key findings obtained during the development of the proposed method to measure input torque have been discussed in Section 5 together with recommendations for future work.

The method presented in this paper could make measuring gearbox torque more cost-effective, facilitating its adoption in

serial wind turbines. This is important since accurate knowledge of the input torque is key to improving gearbox reliability. Furthermore, implementing a torque measurement on each serial wind turbine would permit novel data-driven control strategies,





which can improve drivetrain loading. Having an accurate measurement of the input torque throughout the service history of the gearbox would also enable an improved assessment of the consumed fatigue life of the gearbox components. This knowledge could lead to future design improvements, which would, in turn, lead to higher reliability and lower CoE.

*Data availability.* Due to confidentiality agreements with research collaborators, research data supporting this publication can only be made available to bona fide researchers subject to a non-disclosure agreement. Details of the data and how to request access are available from the 4TU.ResearchData repository at https://doi.org/10.4121/c.5491152.

*Author contributions.* UGS conducted the tests and performed the data analysis presented in the manuscript. All authors provided important input to this research work through discussions, through feedback and by improving the manuscript.

*Competing interests.* The authors declare that they have no conflict of interests.

*Acknowledgements.* We would like to sincerely acknowledge the support of Siemens Gamesa Renewable Energy and TU Delft which made this research possible and the collaboration with Sensing 360 B.V. and DMT GmbH & Co. KG.



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
