# Peer review of "Input Torque Measurements for Wind Turbine Gearboxes Using Fiber Optical Strain Sensors"

_Wind Energy Science, 2021_

## Referee Comment (RC2)

This manuscript presented a high-quality research with a new approach for measurement of the gearbox input torque. Different from the traditional way using strain gauge on the rotating shaft, optical FBG strain sensors were used here on the stationary gearbox housing. Additionally, the alternating part of the signal (due to gear meshing) was used for the torque measurement instead of the absolute value of the signal. As a result, the method can overcome some of the major difficulties in the traditional ways of torque measurement and can be seen as a promising method - at least for certain applications like condition monitoring. However, the method still has to be quantitively validated and certain concerns (from my point of view) still need to be clarified in the **future**:

– The influence of the non-torque load
– Strain distribution along the length of the tooth
– The stability of the calibration under different load cases and over time
– The speed of data processing and possibility of real-time high-frequency measurement
– Time expense in the instrumentation of FBG sensors and cost of interrogator (share use with other measurements?)

The authors have managed to apply the method on a real-size wind turbine gearbox in a professional test campaign and properly analysed the measured data with novel processing methods. Generally, the manuscript made scientific contributions in the measurement and processing methods while proposed a promising method for the industrial application at the same time.

Regarding the content of the manuscript, I would like to give my comments in **the following points**:

1. Section2.3. The position and direction of the FBG sensors.
   After reading the manuscript I would assume that the FBG sensors are installed in the mid-line of the ring gear outer surface and that the sensors are measuring strain in the tangential direction. But since this method was new to me, it took me some time to wonder where exactly and what kind of strains are measured. It would be helpful to add the information explicitly in the text or under one of the figures.

2. Section 2.3 and Section 2.5. Temperature.
   More clarification is needed regarding the temperature influences. As is known that FBG sensors are prone to influences of temperature changes, it is expected from a reader with measurement background that the signal with measured temperature somehow compensated. In the manuscript, it is only mentioned that two temperature sensors are available. After reading through, several questions arise:
   A) are two temperature sensors enough?
   B) at which positions are the two temperature sensors located?
   C) it looks that the temperature measurements are not used in the post processing, the signal was simply detrended instead. why?

   With the given information I could only imagine: the methods focus only on the changes in the strain signals and therefore are not sensitive to the long-term drift caused by temperature change? A temperature non-uniform distribution along the circumference causes probably only a 1P contribution to the Coleman method and very small influence on the 5P, which is used for the torque measurement? Can you please give more words on the topic of temperature influence?

3. Section3.2. Difference in linearity.
   For discussion: could it be that the position of the FBG sensor relative to the nearby tooth/teeth has some contribution to the difference in the slops? The sensor directly under a tooth will likely experience different strain progress as the sensor laying under the middle of

two adjacent teeth. Otherwise, the distribution of strain along the length of the tooth could theoretically also play a part.

4.  Page 20 line 333. Correlation of the phase angle with the torque.
    It appears to me that the phase angle describes how the strain are distributed along the circumference upon a certain point of time and is mostly dependent on the azimuth angle of the planet carrier. Is there theoretical basis supporting the correlation?

5.  Figure 22 and 23. Comparison with HSS torque possible?
    I'm wondering why the HSS torque was not included in the comparison figures. A direct thought will be to compare the measurement with the calibration reference, which is the mean value of the HSS torque times gearbox ratio. Despite of the dynamics inside the gearboxes, such a comparison can still show the low-frequency behaviours of the measurement. Is there a reason that the HSS torque are not drawn in the figures for comparison?

The followings are comments for the text and language, only as recommendations.

6.  Line 61. "3-stage" gearbox.

7.  Line 102&136. "on" the surface.

8.  Line 114. "input load excitation"

9.  Line 115.  with the main axis "horizontal"

10. Line 140. the interrogator "sends" a full…

11. Line 141. the word synchronize strongly implies the time sync. A better word should be possible here. For example, "associate"?

12. Line 158. First, a "test" with a …

13. Line 159. The sentence "In both variable …" should be place in front of sentence "First, a test with …"

14. Line 240. "As can be seen in …"

15. Line 414. "On one hand, …"

---

## Author Comment (AC1)

**Response to Referee Comment #1, wes-2021-69**

**RC1**: 'Comment on wes-2021-69', Jonathan Keller, 10 Sep 2021
**Citation**: https://doi.org/10.5194/wes-2021-69-RC1

Thank you for your positive feedback and constructive comments, which we consider very helpful and will improve the quality of our paper. All comments have been addressed below and the manuscript will be revised accordingly.

**General comments:**

The article discusses an interesting and useful application of fiber Bragg grating sensors for torque-measurement and planetary load-sharing characteristics in wind turbine gearboxes, including full—scale verification testing. In general, I like the article and idea and have made some specific comments. The following comments may be personal preference, but I think some of the detailed descriptions could be shortened, especially in sections 3 and 4, and figures omitted to produce a more succinct article. In a similar fashion, I'd recommend a greater focus or structuring on the two uses and results of this technology (torque measurement and load-sharing) rather than the two methods (peak-to-peak and coordinate transformation). The Conclusions I think could be better written to reflect some of the same things in the Abstract, which is well written. After having read the paper, I am wondering if the title should reflect a greater focus on "Methods", like "Methods for Measuring Input Torque of Wind Turbine Gearboxes Using Fiber Optical Strain Sensors"?

We agree with your suggestion that some figures can be omitted and some of the detailed descriptions shortened. In this regard, the manuscript will be revised as detailed below in the specific comments. However, we believe the current structure is more suitable to describe the proposed method while focusing on the torque measurement application, which is the article's main aim.

We have decided to keep the original title in the revised manuscript because in our opinion conveys the article's content. In previous versions of the article, we did have a similar title to the one you suggested, but we replaced it with the current one to make it less wordy.

**Specific comments:**

**In 1 Introduction:**

- The discussion of gearbox dynamics (p. 2 lines 36-43) feels a bit incomplete. The lack of understanding and the importance of gearbox dynamics are

indeed described, but what I think is implied (but left out) is the ability of this particular sensor to measure the dynamic characteristics of the torque at the frequencies necessary for such investigations – especially as described by the "second signal processing procedure" described in the Abstract and elsewhere. An easy way to improve this might be as easy as adding "dynamic" on to the sentence "It is, therefore, highly desirable to be able to measure the dynamic torque from the rotor acting on the gearbox accurately and reliably."

We agree with your recommendation and have added it to the revised version of the manuscript.

- It might be worth referencing recent work by Winergy on their Digital Gearbox (https://www.winergy-group.com/en/DigitalGearboxUseCase). If I understand correctly, this system is envisioned to be installed on operational wind turbines, rather than the more "one-off" systems (Guo and Rosinski) currently referenced. Maybe it is also important to state in the first sentence that "The traditional method to measure gearbox input torque…"

Thank you for this suggestion. We have added this reference as, indeed, it makes the argumentation to need a direct high-frequency torque measurement stronger.

- I think the sentence describing the main contributions of the paper could be rewritten – the existing sentence is a bit "mixed" I think. In summarizing what this paper is, I wrote the following 2 sentences. They are even a more condensed version of what appears in the Abstract, which I think it already well written: "This paper develops a method to measure input torque on wind turbine gearboxes from ring gear strain measured with optical fiber Bragg grating sensors and demonstrates it through full-scale dynamometer testing. The applicability of this method to also determine planet load-sharing characteristics is also explored."

We acknowledge your comment and will rephrase the contributions paragraph in the revised manuscript.

**In 2 Background:**

- In the first sentence of the Background, I think it would be better to say "The primary function of the gearbox is to transfer the power generated…"

We agree and will change "torque" to "power" in the revised manuscript.

- I think it might be worthwhile to say "The radial and tangential components of the mesh force, resulting from the helix angle in most gears, acting from the planets to…"

  Thank you for this suggestion. We have added a clarification in the revised manuscript to explicitly state that the pressure angle in the gears causes the radial and tangential components of the mesh force and that the helix angle causes the axial component.

- Figures 4 through 7 are all interesting but might be hard to see and don't necessarily add much to the paper. If any changes for brevity were needed, I believe any of these could be omitted. This is just an opinion, though. Maybe others find them very valuable. There are 23 total figures in the paper, which does seem like a large amount.

  We agree with your suggestion and have decided to omit Figures 4 and 6 in the revised manuscript. The interested reader can find Figure 4 in the reference provided and Figure 6 can be omitted without losing much information about the surface preparation before installing the fibers. However, we believe Figure 5 is helpful to understand the position of the sensor used for detecting the position of the planet carrier and the arrangement of the planet gears.  We think Figure 7 offers a detailed view of a fiber grating after installation, which we hope some readers without practical experience using optical fiber sensors will find useful.

- I think the sentence in 2.4 should be "First, tests with a linearly increasing torque command."

  We have rephrased this sentence in the revised manuscript.

- It feels like a better title of section 2.5 might be "Data acquisition and vetting" more so than signal processing, as it feels like sections 3 and 4 are the "real" signal processing steps. This could just be a matter of opinion.

  We appreciate this comment and have reworded the section's title to "Data acquisition and signal pre-processing".

**In 3 Torque Estimation:**

- There are misspelled "toque" at 2 places in 3.2.

  Corrected in the revised manuscript.

- I think some condensing of figures between 11 – 15 could occur. This relates to my general comment that the results and discussion are a bit "method heavy" rather than "results-focused".

Figure 11 has been omitted in the revised manuscript.

**In 4 Torque Estimation using a coordinate transformation**

- Figures 17 – 20 are barely discussed. This relates to my general comment that the results and discussion are a bit "method heavy" rather than "results-focused".

  We believe these figures are helpful to understand the data manipulations presented in Section 4 and have decided to keep them.

**In 5 Discussion:**

- Figure 21 mis-spells "Weight" in the upper portion.

  Corrected in the revised manuscript.

- In terms of reducing figures as commented earlier, Figures 22 and 23 (right) are not discussed (or barely mentioned) in the text and are probably not needed.

  We believe these figures are the central element of the discussion/results section and have decided to keep them. We will add further clarification in the revised manuscript to emphasize their importance.

- I am left with the impression that the coordinate transformation consistently yields better results than the peak-to-peak method. Is this correct, or are there pros and cons to each? If the coordinate transfer method is indeed "better", then I have to wonder the value (in a journal article) of even discussing the peak-to-peak method any more than very briefly. This could be a matter of opinion as it really only relates to the overall length of the paper. Then again, having looked at things more – can load-sharing only be estimated with the peak-to-peak method?

  We appreciate this comment but believe the peak-to-peak procedure does have its pros and provides valuable information, mainly to validate advanced models and to enable the investigation of the load-sharing behavior between the planets. We will revise the discussion section to emphasize these points.

**In 6 Conclusions:**

- Fully summarizing key points I think could be very helpful. For example, instead of "…optical fiber strain sensors were used because of their advantages over more conventional electrical strain gauges", I would suggest to say something like "…optical fiber strain sensors were used because of their higher signal-to-noise ratio, immunity to electromagnetic interference, and faster installation compared to conventional electrical strain gauges" (or whatever the authors feel appropriate). I mention this because by the time I read the Conclusions I couldn't remember what the advantages are, so I had to search back through the document to find the explanation. In a similar fashion, I suggest better summarizing "The key findings obtained during the development of the proposed method to measure input torque have been discussed…together with recommendations for future work." Please state them here!

  We find this suggestion very valuable and will revise the Conclusions section to include explicitly the main advantages of fiber optical sensors, our study's key findings, and the suggestions for future work.

Thank you!

---

## Author Comment (AC2)

**Response to Referee Comments #1, wes-2021-69**

**RC2**: 'Comment on wes-2021-69', Hongkun Zhang, 02 Nov 2021
**Citation**: https://doi.org/10.5194/wes-2021-69-RC2

Thank you for your positive feedback and constructive comments, which we consider helpful in improving the quality of our paper. All comments have been addressed below and the manuscript will be revised accordingly.

This manuscript presented a high-quality research with a new approach for measurement of the gearbox input torque. Different from the traditional way using strain gauge on the rotating shaft, optical FBG strain sensors were used here on the stationary gearbox housing. Additionally, the alternating part of the signal (due to gear meshing) was used for the torque measurement instead of the absolute value of the signal. As a result, the method can overcome some of the major difficulties in the traditional ways of torque measurement and can be seen as a promising method - at least for certain applications like condition monitoring. However, the method still has to be quantitively validated and certain concerns (from my point of view) still need to be clarified in the **future**:

- The influence of the non-torque load

- Strain distribution along the length of the tooth

- The stability of the calibration under different load cases and over time

- The speed of data processing and possibility of real-time high-frequency measurement

- Time expense in the instrumentation of FBG sensors and cost of interrogator (share use with other measurements?)

The authors have managed to apply the method on a real-size wind turbine gearbox in a professional test campaign and properly analysed the measured data with novel processing methods. Generally, the manuscript presented a high-quality research and made scientific contributions in the measurement and processing methods while proposed a promising method for the industrial application at the same time.

We fully agree with the suggested future work and are actively seeking ways to accomplish these research topics:

- Quantitative assessment of the accuracy of the proposed method.
- Test this new method in an environment with non-torque loads, either in a suitable test bench or directly on a wind turbine.
- Study the effect of load distribution along the face width of the gears on the outer surface strain measurements.

- Repeatability of results over time.
- Cost improvements for serial implementation.

Regarding the content of the manuscript, I would like to give my comments in **the following points**:

**1. Section2.3.  The position and direction of the FBG sensors.**

After reading the manuscript I would assume that the FBG sensors are installed in the mid-line of the ring gear outer surface and that the sensors are measuring strain in the tangential direction. But since this method was new to me, it took me some time to wonder where exactly and what kind of strains are measured. It would be helpful to add the information explicitly in the text or under one of the figures.

 Your assumptions are correct. The fibers were placed in the middle sections of the ring gear in the axial direction. The fibers run tangentially to this section, covering the complete revolution along the outer perimeter of the ring gear. The FBGs measure the deformation of the outer surface of the ring gear in the tangential direction of Figure 2.

We acknowledge your comment and will clarify this point in the revised manuscript.

**2. Section 2.3 and Section 2.5. Temperature.**

I think more clarification is needed regarding to temperature influences. As is known that FBG sensors are prone to influences of temperature changes, it is expected from a reader with measurement background that the signal with measured temperature somehow compensated. In the manuscript, it is only mentioned that two temperature sensors are available. After reading through, several questions arise:

- A) are two temperature sensors enough?
- B) at which positions are the two temperature sensors located?
- C) it looks that the temperature measurements are not used in the post processing, the signal was simply detrended instead. why?

With the given information I could only imagine: the methods focus only on the changes in the strain signals and therefore are not sensitive to the long-term drift caused by temperature change? A temperature non-uniform distribution along the circumference causes probably only a 1P contribution to the Coleman method and very small influence on the 5P, which is used for the torque measurement? Can you please give more words on the topic of temperature influence?

Again, your assumptions are correct, and we agree that more clarification on this topic will be beneficial. We will modify the manuscript to add further explanations on how we dealt with temperature effects.

The wavelength reflected by the FBGs shift both due to temperature changes in the grating and due to strain (Section 2.3.). The FBGs we used in our study have a theoretical sensitivity to strain of 1.19 pm wavelength shift per $\mu$m/m strain and 27.9 pm/$°C$ .

Coming back to your specific questions:

A) In the fiber setup explained in Section 2.3. we describe the arrangement of the FBGs. Each fiber was manufactured with 14 gratings to give a total of 56gratings. We decided to use 54 of these gratings for strain measurements in the points described in Figure 2.
   With the two remaining gratings we attempted a temperature measurement to gain information about the temperature in the ring gear and the potential difference in temperature of different locations. A small tube was placed around the fiber to prevent the effect of strain on the reflected wavelength. However, the installation was not completely successful, and we could not use the data.
B) These two gratings were placed in the vicinity of the strain sensors S02 and S29. Our reasoning for choosing these locations was that the lower part of the ring gear immersed in the oil bath could have a different temperature to the top.
C) We did indeed detrend the signal to remove the long-term drift caused by temperature, and we assigned the remaining alternating signal as purely caused by the strain imposed from the meshing of the planets. As explained in Section 2.5. we did not perform real-time analysis of the strain signals and the data was detrended prior to processing.

Your suggestion about the potential effect of a non-uniform temperature distribution on a real-time scenario is very interesting and we will try to address it in future work. All the tests presented in this study were performed under stabilized temperature conditions. The effects of transient operating conditions with large temperature gradients need to be studied.

**3. Section3.2. Difference in linearity.**

For discussion: could it be that the position of the FBG sensor relative to the nearby tooth/teeth has some contribution to the difference in the slops? The sensor directly under a tooth will likely experience different strain progress as the sensor laying

under the middle of two adjacent teeth. Otherwise, the distribution of strain along the length of the tooth could theoretically also play a part.

We agree that the position of the strain sensors relative to the gear teeth, which does change for the different angular positions, and the load distribution along the face width may influence the slope of the strain to torque relationship. However, we failed to understand why some sensors have a more linear behavior than others.

**4. Page 20, line 333. Correlation of the phase angle with the torque.**

It appears to me that the phase angle describes how the strain are distributed along the circumference upon a certain point of time and is mostly dependent on the azimuth angle of the planet carrier. Is there theoretical basis supporting the correlation?

Our theoretical assumption to expect a change in the phase angle is based on the torsional deformation of the planet carrier caused by the torque. The azimuth of the shaft was given by the once-per-revolution pulse from the inductive sensor placed on the rotor side flange of the planet carrier. The phase angle depends on the azimuth angle, but we were expecting to observe a phase delay related to the amount of twist in the carrier, which would increase with the torque.

**5. Figure 22 and 23. Comparison with HSS torque possible?**

I'm wondering why the HSS torque was not included in the comparison figures. A direct thought will be to compare the measurement with the calibration reference, which is the mean value of the HSS torque times gearbox ratio. Despite of the dynamics inside the gearboxes, such a comparison can still show the low-frequency behaviours of the measurement. Is there a reason that the HSS torque are not drawn in the figures for comparison?

For the calibration phase, we used the signals precisely as you describe. We multiplied the torque measured by each transducer at the high-speed shaft by the gearbox ratio and then computed the mean value of the two gearboxes. This does not capture the dynamics of the gearboxes and we felt we could not use this computed signal to compare the high-frequency input torque estimates. Adding this reference torque value created graphs that were less clear and more difficult to read, so we decided to compare the two estimates without the reference. However, we did not consider filtering the high-frequency content to show the low-frequency behavior. We will consider this suggestion for the revised manuscript.

The followings are comments for the text and language, only as recommendations.

6. Line 61. "3-stage" gearbox.

Corrected in the revised manuscript.

7. Line 102&136. "on" the surface.

Corrected in the revised manuscript.

8. Line 114. "input load excitation"

Added in the revised manuscript.

9. Line 115.  with the main axis "horizontal"

Rephased in the revised manuscript.

10. Line 140. the interrogator "sends" a full...

Changed in the revised manuscript.

11. Line 141. the word synchronize strongly implies the time sync. A better word should be possible here. For example, "associate"?

Changed in the revised manuscript.

12. Line 158. First, a "test" with a ...

Rephased in the revised manuscript.

13. Line 159. The sentence "In both variable ..." should be place in front of sentence "First, a test with ..."

Rephased in the revised manuscript.

14. Line 240. "As can be seen in ..."

Rephased in the revised manuscript.

15. Line 414. "On one hand, ..."

Rephased in the revised manuscript.

Thank you!